# Positive Effect of Antagonistic Additives on the Homogeneous Catalytic Etherification Reaction of Glycerol

Taeyoul Han [1] and Je Seung Lee [2,*]

1   Green Carbon Research Center, Korea Research Institute of Chemical Technology (KRICT), Daejeon 34114, Korea; hty8328@krict.re.kr

2   Department of Chemistry and Research Institute of Basic Sciences, Kyung Hee University, 26 Kyungheedae-ro, Dongdaemun-gu, Seoul 02447, Korea

*   Correspondence: leejs70@khu.ac.kr; Tel.: +82-2-961-0458

**Abstract:** Various compounds prepared using glycerol, diglycerol (DG), and triglycerol (TG) have been gaining increasing attention due to their wide range of applications. To increase the yield and selectivity of DG and TG syntheses, previous studies investigated a variety of catalysts with different basicity and variable reaction temperatures. In this study, we introduced additives that act as inhibitors to increase the selectivity of the etherification reaction for DG and TG production and depress the formation of higher oligomers by moderating the activity of the catalyst. By adding weakly acidic alkali metal-based inorganic salts ($NaHSO_4$ and $KHSO_4$), the selectivity of DG and TG formation could be enhanced, although the conversion of glycerol decreased due to the reduced activity of catalyst. We found that the decrease in the activity of the catalyst caused by the additives could be recovered and that side reactions were reduced if the reaction was carried out at an increased temperature of 280 °C and if the reaction time was shortened to 2 h to suppress the formation of oligomers. The dependence of the reaction on the amount of the additive, the reaction time, and the reaction temperature was investigated to elucidate the role of the additive.

**Keywords:** glycerol; diglycerol; triglycerol; proton donor; additive; homogeneous catalysis

## 1. Introduction

Biodiesel, the methyl ester of various natural fatty acids, is prepared by the transesterification reaction of triglycerides obtained from vegetable oils and methanol using acid or base catalysts and producing ~10% glycerol as a byproduct [1–3]. In 2008, biodiesel and glycerol were produced globally in quantities of 10.1 million and 1.1 million tons, respectively; however, the consumption of glycerol was only 0.6 million tons [4]. For this reason, various studies of the catalytic conversion of glycerol to high value-added commodity chemicals including acrolein [5,6], allyl alcohol [7–9], epichlorohydrine [10], glycerol carbonate [11,12], glycidol [13,14], acetins [15–18], and condensed glycerols such as diglycerol (DG), triglycerol (TG), and polyglycerols [19–23] have been proposed. Among these materials, DG and TG prepared by the etherification reaction of glycerol have been used widely in applications including pharmaceuticals, food additives, cosmetics, lubricants, plasticizers, stabilizers, dispersants, and wetting agents [24–27].

DG and TG are conventionally produced by the reaction of glycerol with glycidol or epichlorohydrin with high selectivity and yield. However, this method suffers from several drawbacks including expensive starting materials and increased post-treatment costs due to the generation of hydrochloric acid (HCl) as a by-product [28,29]. Although the selective etherification of glycerol can proceed under both acidic and basic conditions, basic conditions are preferable due to the production of undesirable by-products such as acrolein under acidic conditions [30,31]. For this reason, several attempts to directly convert glycerol into DG and TG by the selective etherification of glycerol in the presence of strong basic catalysts such as KOH, CsOH, $Li_2CO_3$, $Na_2CO_3$, NaOH, $K_2CO_3$, Cs-MCM41,

metal oxides, etc. [1,23,32–35] have been made. DG is produced by the condensation between two hydroxyl groups on two glycerol molecules, generating a water molecule as a by-product [21,34]. The catalytic etherification reaction between DG and another glycerol molecule yields TG. Further etherification reactions of glycerol and/or products generate oligomers or PGs. Although the etherification of glycerol is a simple condensation reaction, controlling the selectivity is an onerous task because each of three hydroxyl groups on glycerol may participate in condensation reactions [36,37]. Therefore, the yield and selectivity of DG and TG are typically poor in the presence of strong bases due to the production of various by-products including oligomers, polymers, and cyclic compounds [21,31,32].

As shown in Table 1, etherification of glycerol using 1.74 mol% of $Na_2CO_3$ as a catalyst converted 96% of glycerol and exhibited selectivities of 24 and 35% for DG and TG, respectively, at 260 °C and after 8 h of reaction [38]. As can be seen from the results using $Na_2CO_3$ and $NaHCO_3$, increased reaction temperature and time resulted in enhanced conversion of glycerol but decreased selectivity of DG and TG due to decreased control of the reaction and further condensation reactions of DG and/or TG under more severe conditions [19,31,38–41]. Although the reported highest selectivity of DG is 60% at 240 °C and 8 h, relatively large amounts of NaOH (4.60 mol%) are needed, and only 63% of glycerol is converted [39]. Most of the existing methods for the etherification of glycerol have the disadvantages of long reaction times (6–8 h) and generating large amounts of oligomers. Unlike DG, TG, and unreacted glycerol, which can be separated from the reaction mixture by distillation under reduced pressure and reused, oligomeric by-products larger than the tetramer of glycerol are difficult to separate or utilize as they comprise non-volatile mixtures of mixed molecular weights. Thus, there is a need to suppress the formation of oligomers and other by-products at high reaction temperatures and long reaction times.

**Table 1.** Etherification reactions of glycerol with basic catalysts.

| Catalyst | Amount of Catalyst (mol%) | Reaction Time (h) | Conversion of Glycerol (%) | Selectivity (%) | | | Reference |
| --- | --- | --- | --- | --- | --- | --- | --- |
| | | | | DG | TG | Others | |
| $Na_2CO_3$ [1] | 1.74 | 8 | 96 | 24 | 35 | 41 | [38] |
| NaHCO3 [1] | 0.22 | 8 | 75 | 27 | 12 | 61 | [31] |
| NaOH [2] | 4.60 | 8 | 63 | 60 | 32 | 7 | [39] |
| NaOH [1] | 0.50 | 6 | 83.8 | 24.5 | 19.5 | 56 | [21] |
| KOH [1] | 0.50 | 6 | 91.9 | 18.9 | 14.5 | 66.6 | [21] |
| NaOAc [1] | 0.50 | 6 | 72.8 | 38.7 | 31.9 | 29.4 | [21] |
| NaOAc [3] | 0.50 | 6 | 59.9 | 53.6 | 26.9 | 19.5 | This work |
| NaOAc [4] | 0.50 | 2 | 77.0 | 41.5 | 28.1 | 30.4 | This work |
| NaOAc [5] | 0.50 | 2 | 55.4 | 62.5 | 28.5 | 9.0 | This work |
| KOAc [1] | 0.50 | 6 | 82.6 | 32.8 | 25.1 | 42.1 | [21] |
| KOAc [3] | 0.50 | 6 | 69.6 | 31.5 | 23.5 | 45.0 | This work |
| KOAc [4] | 0.50 | 2 | 88.5 | 24.3 | 21.7 | 54.0 | This work |
| KOAc [6] | 0.50 | 2 | 62.8 | 52.3 | 28.7 | 19.0 | This work |

[1] Reaction temperature = 260 °C. [2] Reaction temperature = 240 °C. [3] Reaction temperature = 260 °C, 3 mol% of hexanoic acid was added as a proton donor. [4] Reaction temperature = 280 °C. [5] Reaction temperature = 280 °C, 0.37 mol% of $NaHSO_4$ was added as a proton donor. [6] Reaction temperature = 280 °C, 0.37 mol% of $KHSO_4$ was added as a proton donor.

Previously, we reported the etherification reaction of glycerol using basic acetate salts, MOAc (M = Li, Na or K), to produce the DG and TG [21]. By using a weakly basic acetate salt as the catalyst, increased selectivities of DG and TG could be achieved compared to previous reports using strong bases. However, there were limitations for increased selectivity and yields of DG and TG using a homogeneous catalytic reaction and only controlling the basicity of the catalyst, the reaction temperature, and the reaction time. As an extension of our previous work, we herein investigated the effects of additives, $MHSO_4$ (M = Na or K), as proton donors and their positive antagonistic role in the etherification of glycerol. To the best of our knowledge, this is the first study to increase the selectivities and

yields of DG and TG using inhibitors that lower the activity of the catalysts as additives by suppressing the formation of oligomers in the catalytic etherification of glycerol.

## 2. Results and Discussion

Selective etherification of glycerol is an alluring reaction pathway because the products can be directly converted into useful chemicals. In this work, the conversion of glycerol was carried out in the presence of a basic catalyst using a Dean–Stark apparatus to remove water generated during the reaction.

The results of the etherification reaction of glycerol using basic catalysts, including previously reported results, are summarized in Table 1. It can be seen that under the conditions of strong catalyst basicity, long reaction times, and high reaction temperatures, the conversion of glycerol increases, but the selectivities of DG and TG decrease. For example, as shown in Table 1, under the same reaction conditions, the conversion of glycerol in the presence of NaOAc and KOAc was 72.8 and 82.6%, respectively. Meanwhile, the selectivities of DG and TG in the presence of NaOAc and KOAc were 38.7, 31.9 and 32.8, 25.1%, respectively. This is because DG and TG produced by the etherification of glycerol further reacted to form the oligomers. Therefore, it is necessary to inhibit the formation of oligomers by controlling the activity of the catalyst to increase the selectivities of DG and TG.

By adding an appropriate proton donor, it is expected that the reaction rate will be slowed due to the protonation of glycerol, DG, and TG. Under the same reaction conditions, the proton donor was added to inhibit oligomer formation. Since it was determined to be advantageous to add a weak acid having a relatively high boiling point, hexanoic acid (which has a relatively high boiling point 205.8 °C) and weak acidity ($pK_a$ = 4.88) was added as an additive to decrease the reactivity of catalysts. As shown in Table 1 and Figure 1, by adding hexanoic acid as a proton donor, the conversion of glycerol was suppressed as expected, but the formation of oligomers did not significantly decrease. Based on a comparison of Figure 1a,b, in the presence of NaOAc as a catalyst, the addition of hexanoic acid effectively inhibited the formation of oligomers at the beginning of the reaction, but the formation of oligomers became remarkable and the selectivity of DG decreased after 4 h. These results indicate that the hexanoic acid played a role in lowering the activity of the catalyst at the beginning of the reaction. As the reaction proceeded, however, some amount of the hexanoic acid was lost by evaporation with water generated as a by-product due to the lower boiling point of hexanoic acid than the reaction temperature. As a result, the formation of oligomers increased as the further reactions of DG and TG were accelerated after the reaction time of 4 h (Figure 1b). In the case that KOAc was used as a catalyst, hexanoic acid did not lower the catalytic activity. Although the conversion of glycerin was decreased slightly by the addition of hexanoic acid, the formation of oligomers was not inhibited efficiently even early in the reaction (Figure 1c,d). This may be due to the stronger basicity of KOAc and the weak acidity of hexanoic acid.

To overcome the volatility and weak acidity of hexanoic acid, a non-volatile, inorganic salt (NaHSO$_4$ or KHSO$_4$, $pK_a$ = 1.99) was added as an additive and its effects were examined. Additionally, based on the results of previous studies showing that the production of oligomers increased as the reaction time increased [21], a relatively short reaction time of 2 h at 280 °C was used to reduce the formation of oligomers. As can be seen in Table 1, both NaOAc and KOAc catalysts resulted in higher glycerol conversion rates under the reaction conditions of 280 °C and 2 h than at 260 °C and 6 h with the bisulfate additive. In both acetate salts, however, the selectivities of DG and TG were similar or lower and the yield of oligomer increased relatively. By adding 0.37 mol% of NaHSO$_4$ or KHSO$_4$ to the reaction mixture as a proton donor and allowing the reaction to proceed at 280 °C for 2 h, the conversion of glycerol was decreased as in the case of hexanoic acid, but the selectivity of DG and TG increased (from 41.5 and 28.1% to 62.5 and 28.5% for NaOAc; from 24.3 and 21.7% to 52.3 and 28.7%, respectively) and the formation of oligomers was suppressed significantly (from 30.4 to 9.0% for NaOAc; from 54.0 to 19.0% for KOAc, respectively). These

results, as expected, appear to be due to the fact that the non-volatile additive remained in the reaction mixture during the reaction and suppressed the formation of oligomers by effectively inhibiting the activity of the catalyst.

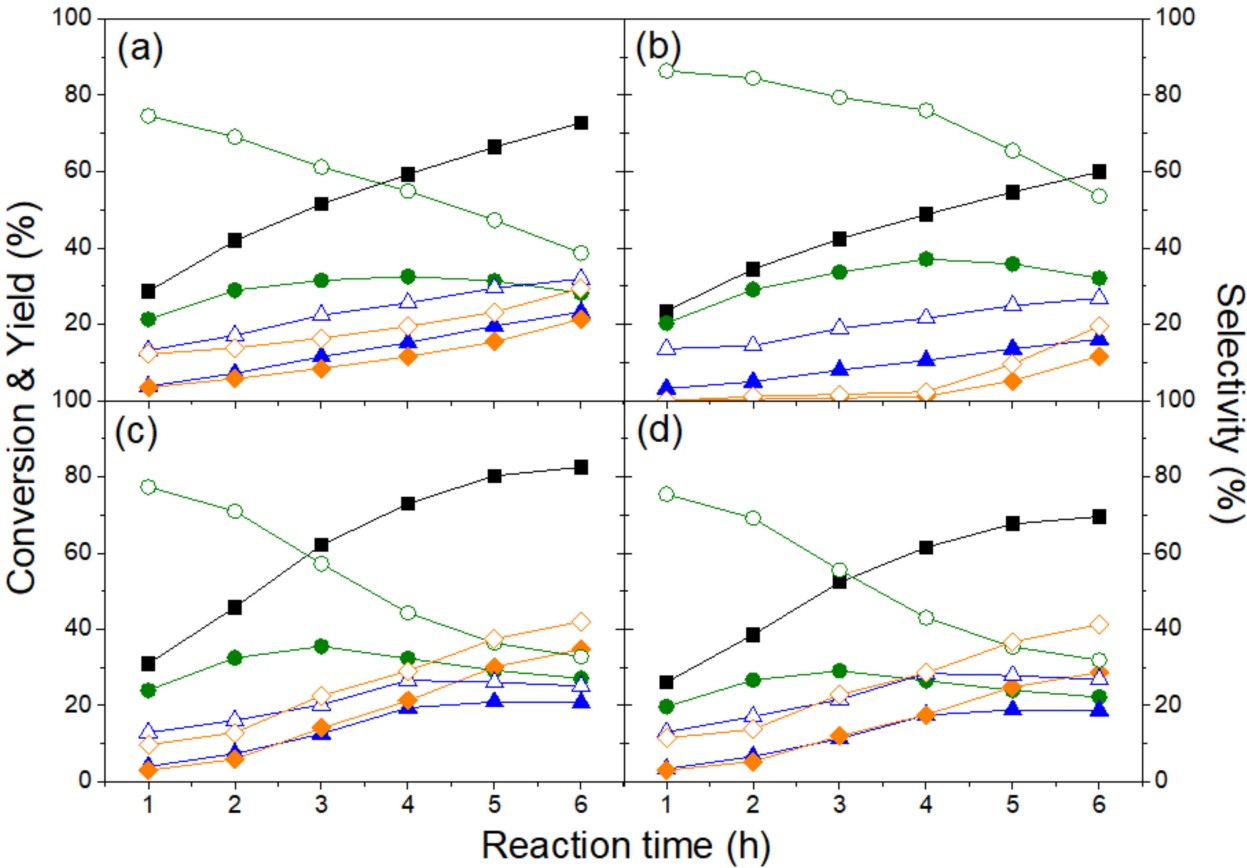

**Figure 1.** Influence of the reaction time on the etherification of glycerol in the presence of (**a**) NaOAc, (**b**) NaOAc + hexanoic acid, (**c**) KOAc, and (**d**) KOAc + hexanoic acid (-■-: conversion of glycerol, -●-: yield of DG, -▲-: yield of TG, -◆-: yield of others -○-: selectivity of DG, -△-: selectivity of TG, -◇-: selectivity of others, the amount of catalyst: 0.5 mol%, the amount of hexanoic acid: 3 mol% and reaction temperature: 260 °C).

Figure 2a–c shows the results of the etherification reaction using NaOAc as a catalyst with variable amounts of $NaHSO_4$ as an additive at 270, 280, and 290 °C, respectively. The amount of NaOAc was 0.5 mol% with respect to glycerol, and the reaction time was 2 h. As the amount of $NaHSO_4$ increased, the conversion of glycerol decreased; however, the sum of the selectivities of DG and TG were significantly improved (up to 91%) in the presence of 0.37 mol% of $NaHSO_4$ due to the decreased formation of oligomers. These results indicate that the activity of the catalyst was effectively suppressed by the proton donor, thereby impeding further reactions of DG and TG, which produce oligomers. Similarly, as shown in Figure 2d–f, in the presence of 0.5 mol% of the KOAc catalyst, the increasing $KHSO_4$ concentration decreased the conversion of glycerol and formation of oligomers while increasing the yields and selectivities of DG and TG. The yields and selectivities of DG and TG in the presence of K salts were lower than those of Na salts due to the stronger basicity of K salts than Na salts.

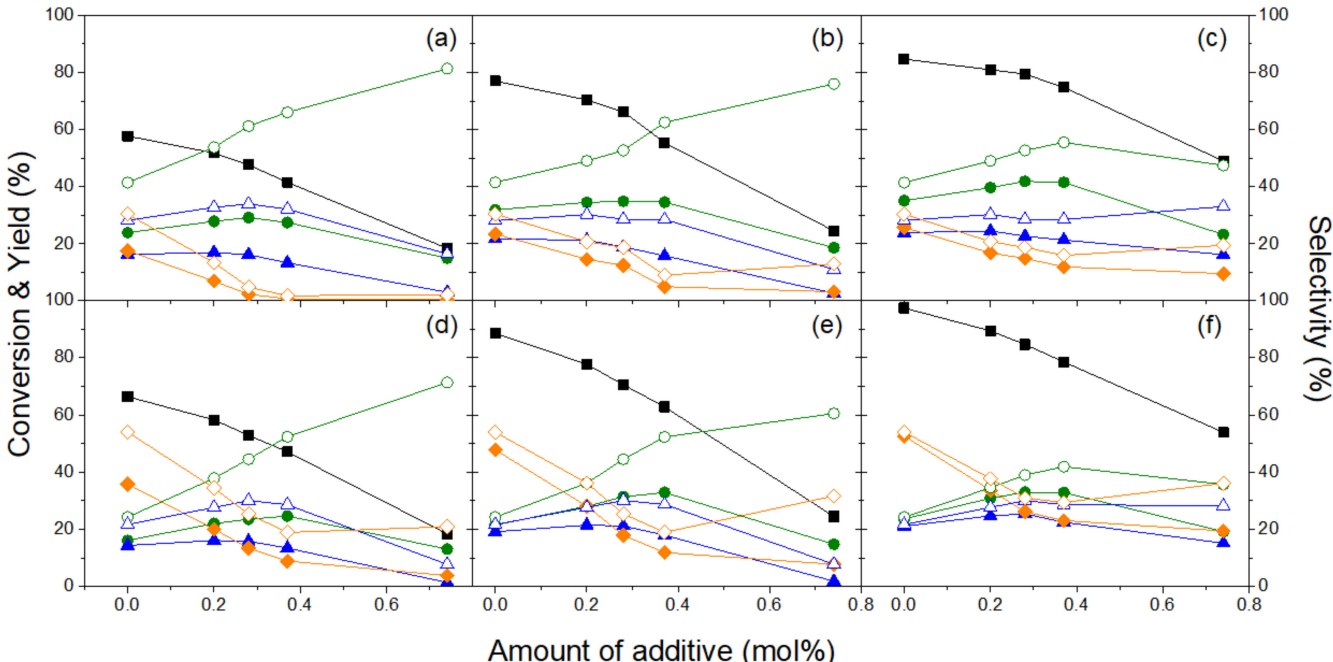

**Figure 2.** The effect of additive concentration on the etherification of glycerol in the presence of NaOAc (0.5 mol%) at (**a**) 270 °C, (**b**) 280 °C and (**c**) 290 °C and KOAc (0.5 mol%) at (**d**) 270 °C, (**e**) 280 °C, and (**f**) 290 °C with variable amounts of NaHSO$_4$ and KHSO$_4$ as an additive (-■-: conversion of glycerol, -●-: yield of DG, -▲-: yield of TG, -◆-: yield of others -○-: selectivity of DG, -△-: selectivity of TG, -◇-: selectivity of others, the reaction time: 2 h).

Figure 3 shows the results of etherification reactions carried out for 2 h at variable temperatures in the presence of NaOAc and NaHSO$_4$ or KOAc and KHSO$_4$ (0.5 and 0.37 mol%, respectively). As the temperature increased, the conversion of glycerol increased and the yields and selectivities of DG and TG showed optimal values at 280 °C; 0.37 mol% of NaHSO$_4$ in the presence of 0.5 mol% of NaOAc showed good selectivities of DG and TG (62.6 and 28.5%) and yields of 34.6 and 15.8%, respectively, at 280 °C. At 290 °C, however, the yields and selectivities of DG and TG decreased due to the formation of oligomers, indicating that the catalytic activities of both K$^+$ and Na$^+$ salts increased at high temperature.

In the presence of 0.5 mol% NaOAc or KOAc and 0.37 mol% NaHSO$_4$ or KHSO$_4$, the etherification reaction was carried out at 280 °C with variable reaction times in the range of 1–3 h. As the reaction time increased, both the conversion of glycerol and the yield of oligomers increased. The yields of DG and TG slightly increased up to 2 h and then decreased, while the selectivity of DG consistently decreased due to the DG transforming into the TG and/or oligomers as the reaction time increased (Figure 4).

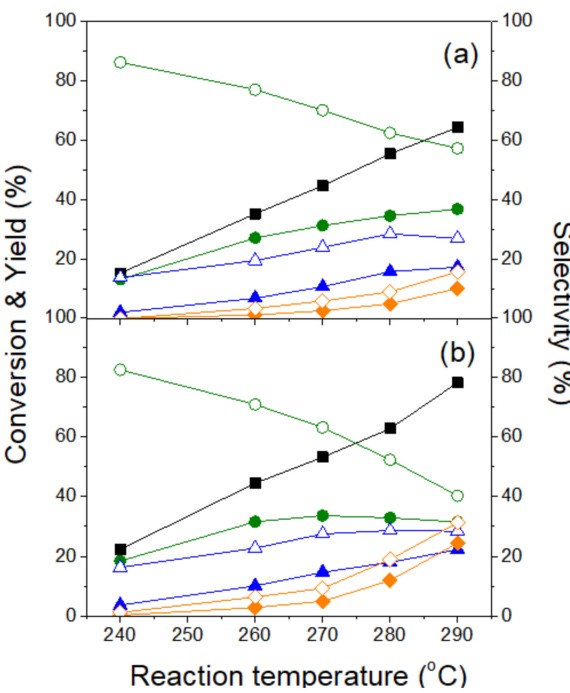

**Figure 3.** Influence of the reaction temperature on the etherification of glycerol in the presence of (**a**) NaOAc and (**b**) KOAc with $NaHSO_4$ and $KHSO_4$, respectively (-■-: conversion of glycerol, -●-: yield of DG, -▲-: yield of TG, -◆-: yield of others -○-: selectivity of DG, -△-: selectivity of TG, -◇-: selectivity of others, the amount of catalyst: 0.5 mol%, the amount of additive: 0.37 mol% and the reaction time: 2 h).

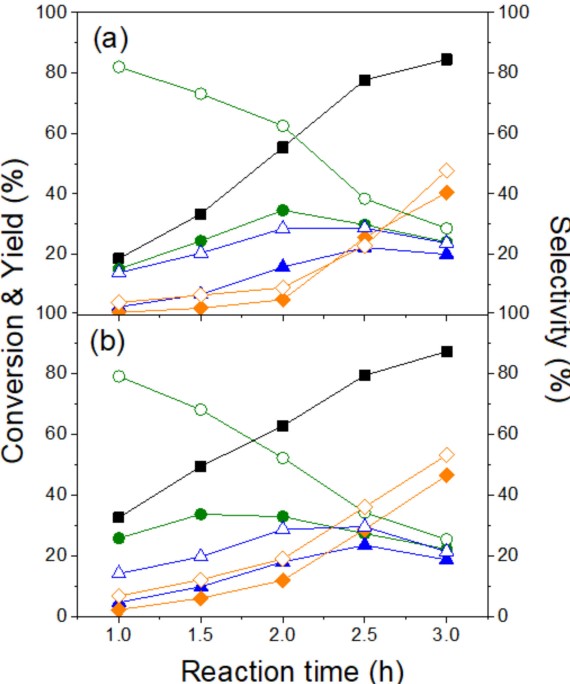

**Figure 4.** The influence of the reaction time on the etherification reaction of glycerol in the presence of (**a**) NaOAc and (**b**) KOAc with $NaHSO_4$ and $KHSO_4$, respectively (-■-: conversion of glycerol, -●-: yield of DG, -▲-: yield of TG, -◆-: yield of others -○-: selectivity of DG, -△-: selectivity of TG, -◇-: selectivity of others, the amount of catalyst: 0.5 mol%, the amount of additive: 0.37 mol% and the reaction temperature: 280 °C).

## 3. Materials and Methods

### 3.1. Materials

Glycerol was purchased from DAEJUNG Chemicals (Seoul, Korea). Sodium acetate (NaOAc), sodium bisulfate ($NaHSO_4$), potassium acetate (KOAc), and potassium bisulfate ($KHSO_4$) were obtained from Sigma–Aldrich Co. (Yongin, Korea). $H_2O$ (HPLC grade) was obtained from J. T. Baker Co. (Seoul, Korea). All the chemicals were used as obtained without further purification.

### 3.2. Etherification Reaction of Glycerol

Glycerol was etherified using 0.5 mol% of catalyst (MOAc) and optionally 0.37 mol% of $MHSO_4$ (M = Na or K) as a proton donor in the temperature range of 260–290 °C under ambient pressure. Glycerol (30 g), the catalyst, and optionally the proton donor were put into a two-necked round bottomed flask (150 mL) that was placed on a heating mantle equipped with a temperature controller. A Dean–Stark apparatus was used to remove the water generated during the reaction.

### 3.3. Analysis of the Reaction Mixtures

Analysis of the reaction mixture was carried out using high-performance liquid chromatography (1220 Infinity I, Agilent Company, Santa Clara, CA, USA) using an Aminex HPX-87H (Bio-Rad Laboratories Inc., Hercules, CA, USA) column and RI (reflective index) detector; 5 mM of $H_2SO_4$ aqueous solution was used as mobile phase and the flow rate was 0.6 mL min$^{-1}$. The obtained signals of each product were calibrated using reference samples with known concentration.

## 4. Conclusions

In order to control the catalytic activity of NaOAc or KOAc used as catalysts for the etherification of glycerol, the additives, $MHSO_4$ (M = Na or K), capable of acting as proton donors, were added to investigate the characteristics of the reaction. The conversion of glycerol and the yield and selectivity of DG, TG, and oligomeric compounds were controlled by varying the amount of additive, reaction time, and reaction temperature; 0.37 mol% of $NaHSO_4$ in the presence of 0.5 mol% of NaOAc showed excellent catalytic performance with the selectivities of DG and TG reaching 62.6 and 28.5%, respectively, at 280 °C and 2 h. Therefore, the selectivities and yields of DG and TG were increased, and the formation of oligomers was suppressed effectively by introducing the additives capable of moderating the activity of the catalyst while using a high reaction temperature and a short reaction time.

**Author Contributions:** Conceptualization, J.S.L.; methodology, J.S.L.; formal analysis, T.H.; investigation, T.H.; data curation, T.H.; writing—original draft preparation, T.H.; writing—review and editing, J.S.L.; visualization, T.H.; supervision, J.S.L.; project administration, J.S.L.; funding acquisition, J.S.L. Both authors have read and agreed to the published version of the manuscript.

**Funding:** This research was funded by the National Research Foundation of Korea (NRF), grant number: 2018R1D1A1B07050522.

**Conflicts of Interest:** The authors declare no conflict of interest.

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
