# Peer review of "Positive Effect of Antagonistic Additives on the Homogeneous Catalytic Etherification Reaction of Glycerol"

_catalysts, doi:10.3390/catal11081000_

Round 1
Reviewer 1 Report
The present paper entitled “Positive effect of antagonistic additives on the homogeneous catalytic etherification reaction of glycerol" by the authors Taeyoul Han and Je Seung Lee, describes the addition of additives to increase the selectivity of diglycerol and triglycerol, temperature and time were some of the parameters studied for this process.
After careful evaluation of the manuscript, I consider that it would typically appeal to the readers of Catalysts but there are some questions to the paper:
Q1: the introduction is well written but it looks that the review presented in the introduction is a little out of date. In 34 references the more recent is some work of one of the authors and it is from 2018 and the rest are all before 2013, there is nothing recent?
If the authors did a simple search with glycerol+etherification+diglycerol or triglycerol in the last 5 years you get more than 15 articles, that are not mentioned in this paper. Even though this paper is studding the effect of the addition of additives the main reaction continuous to the etherifiction of glycerol to diglycerol, and triglycerol, so at least some recent results even though not using the additives should be addressed. In the simple search there are some reviews that can be mentioned.
Q2: Why the reference 31 is not mentioned on Table 1? It is referred in the text line 77-79, when commenting table 1 but it is not there. After that in lines 79-81 some results are referred, and it seems like they belong to this work but I think that they are from reference 17.
Q3: In all the figures the authors refer to selectivity of DG, selectivity of TG, selectivity of others twice, I imagine that one of those should be yield but it is not easy to understand which ones are selectivity and which are yields.
Q4: When analysing the selectivity for DG + TG, table 1, it is possible to observe that similar results were obtained when using MHSO4 or hexanoic acid, even when we see at conversion level. It is true that with hexanoic acid it needed 6 hours of reaction, but on the other hand it was used low temperature. Since hexanoic acid has a low boiling point (205.8ºC) when compared to the temperatures studied (240 – 290 ºC – this should be corrected in the materials and methods), maybe another proton donor similar to this should be examined, but with a higher boiling point in order to increase the temperature to at least 280 ºC, to understand that if the results with this kind of proton donor are better.
Q5: Which is the amount of water in the initial glycerol solution? Without this information is difficult to be certain that the reaction is homogeneous, since MOAc and MHSO4 or MH2PO4 should be difficult to dissolve in pure glycerol. Using a Dean Stark apparatus, the water is separated from the reaction, how can the authors guarantee that the reaction continues to be homogeneous.

Author Response
Thank you for the valuable and constructive comments and suggestions that are of great help in preparing the revised manuscript with improved quality. The changes in manuscript were highlighted using the “Track Changes” function. Please see the attachments.

Reviewer 2 Report
The paper is badly written and needs major revision. It isn't easy to understand the novelty of the present work, as many results that the authors refer to in the discussion have beed already published in their previous work (ref 17). This point has to be clarified in the abstract and in the discussion of the results. A large part of the text from the discussion should be moved to the introduction part of the paper. English needs careful revision. The additives presented in this work decreased conversion and increased selectivity; however, the results without the additives are not shown. Moreover, in this case, the yields should be compared as from the commercial point of view, it is always beneficial to reduce to a minimum the number of used additives. Every additive adds to the cost of the process, so it needs to bring significant improvements to the yield of the targeted products. The small improvement in selectivity does not justify the need for such an additive. This point needs to be clarified.
Further comments are listed below.
The abstract should contain the topic of the research and the main conclusions of the work done. The novelty of the work is not clear from the abstract. Was the addition of these salt studied before?
Line 15-reaction was carried out at
Line 23- The decreasing amount of fossil fuels is not the main reason for researching biofuels. Climate change due to overdependence on fossil fuels is a far greater issue nowadays.
Line 31- conversion of glycerol to acetins should be added
Line 31- remove which could be
Line 49-were seriously poor-Add the references in which the reported yield was poor.
Line 54- even the higher conversion of glycerol? Correct the sentence.
Line 63-“ As reported in the previous…the sentence is too long and difficult to understand. Please divide into two sentances. This info should be moved to the introduction part.
The previous results should be described in the introduction.
Line 74- in 8h reaction time
Line 74-As can be seen from…
Line 77-reaction proceeds
Line 78- the sign of degrees is too low
All information on page two should be included in the introduction part, not in the results. The Table 1 can stay in the results. However, the results obtained in the present work should be discussed first and later on compared to the published data.
Figure 1. Change the caption to Influence of the reaction time on yield of…, conversion of… and selectivity to…The reaction time does not depend on the reaction!
Line 100-slowed down
Line 101-was added to…remove intend
It would be beneficial to add the figure from the ref 17 with the influence of the reaction time on the selectivity and conversion in the absence of hexanoic acid. The discussion includes the data that is not present in the manuscript at the moment. What was the activity of the catalyst at the beginning of the reaction without the addition of hexanoic acid? What is the point of adding hexanoic acid that decreases the conversion and evaporates during the reaction?
From the results in Table 1, it ios clear that the addition of hexanoic acid decreased the conversion of glycerol and did not change the selectivity.
Suppose the additive decreases the conversion but increases the selectivity. In that case, the authors should compae the yields of TG and DG with and without the additives to conclude whether it is worth using this additive or not. Add the comparison of yields of DG and TG at different temperatures with and without the additives.
Line 163-elongated
Author Response

(The authors gave the same response as above.)

Reviewer 3 Report
The paper presents investigation results of the yield and selectivity of diglycerol (DG) and triglycerol (TG), the catalytic activity of acetate salts (NaOAc and KOAc) used in the etherification of glycerol controlled by adding an additive. By the addition of weakly acidic alkali metal-based inorganic salts (NaHSO4 and KHSO4), the selectivity of DG and TG increased. The dependence of the reaction on the amount of the additive, the reaction time and the reaction temperature were investigated to elucidate the role of the additive.
The research area is of interest to both scientists and engineers.
Generally, the quality of this paper is quite good, including quality of the figures and tables. I can recommend it for publication after minor corrections.
I have minor comments:
- In some places, spaces between the number and the sign "℃" should be removed (e.g. 260 ℃);
- The abstract should provide the motivation behind this research. Why is this research important?;
- The list of literature should be enriched with the latest literature.
Author Response

(The authors gave the same response as above.)

Round 2
Reviewer 2 Report
The paper has been significantly improved, however, the minor correction of English is still required prior to its publication.
Author Response
Responses to the Reviewer
Manuscript ID: catalysts-1317635
Title: "Positive effect of antagonistic additives on the homogeneous catalytic etherification reaction of glycerol"
Thank you for the valuable and constructive comments and suggestions that are of great help in preparing the revised manuscript with improved quality. As the reviewer recommended, we polished the English with an aid of our faculty who is native speaker. The changes in manuscript were highlighted using the “Track Changes” function. Please see the attachment.